# Genomic variation and epidemiology of SARS-CoV-2 importation and early circulation in Israel

Neta S. Zuckerman [1]*, Efrat Bucris[1], Yaron Drori[1,2], Oran Erster[1], Danit Sofer[1], Rakefet Pando[1,3], Ella Mendelson[1,2], Orna Mor[1,2�], Michal Mandelboim[1,2�]

**1** Central Virology Laboratory, Ministry of Health, Chaim Sheba Medical Center, Ramat Gan, Israel, **2** School of Public Health, Sackler Faculty of Medicine, Tel-Aviv University, Tel-Aviv, Israel, **3** Israel Center for Disease Control, Israel Ministry of Health, Chaim Sheba Medical Center, Ramat Gan, Israel

☉ These authors contributed equally to this work.

* Neta.Zuckerman@sheba.health.gov.il

**Data Availability Statement:** The sequences are available from NCBI: MW674808 - MW674874, MW672622 - MW672669.

## Abstract

Severe acute respiratory disease coronavirus 2 (SARS-CoV-2) which causes corona virus disease (COVID-19) was first identified in Wuhan, China in December 2019 and has since led to a global pandemic. Importations of SARS-CoV-2 to Israel in late February from multiple countries initiated a rapid outbreak across the country. In this study, SARS-CoV-2 whole genomes were sequenced from 59 imported samples with a recorded country of importation and 101 early circulating samples in February to mid-March 2020 and analyzed to infer clades and mutational patterns with additional sequences identified Israel available in public databases. Recorded importations in February to mid-March, mostly from Europe, led to multiple transmissions in all districts in Israel. Although all SARS-CoV-2 defined clades were imported, clade 20C became the dominating clade in the circulating samples. Identification of novel, frequently altered mutated positions correlating with clade-defining positions provide data for surveillance of this evolving pandemic and spread of specific clades of this virus. SARS-CoV-2 continues to spread and mutate in Israel and across the globe. With economy and travel resuming, surveillance of clades and accumulating mutations is crucial for understanding its evolution and spread patterns and may aid in decision making concerning public health issues.

## Introduction

Severe acute respiratory syndrome coronavirus 2 (SARS-CoV-2) was first identified in Wuhan, China in December 2019 [1] and has since rapidly spread, infecting over 20 million people worldwide to this day. SARS-CoV-2 causes corona viral disease (COVID-19) and was declared a pandemic by the world health organization on March 2020 [2]. Currently, there is no vaccine or approved effective therapeutic treatments [3].

Major SARS-CoV-2 clades have been characterized based on whole viral genome sequencing data, with over 80,000 sequences currently deposited from countries worldwide in the

**Funding:** The authors received no specific funding for this work.

**Competing interests:** the authors have declared that no competing interests exist.

global initiative on sharing all influenza data (GISAID) database [4]. The main nomenclature systems of SARS-CoV-2 clades include Nextstrain, who name a new major clade when it reaches a frequency of 20% globally by using a year-letter genetic clade naming [5], and GISAID, who use the statistical distribution of genome distances in phylogenetic clusters and name the clades by the actual letters of the defining marker mutations of each cluster [4]. According to Nextstrain's nomenclature system [5], five globally circulating SARS-CoV-2 clades are currently defined– 19A (the root clade) and 19B, that originated in Asia and are still widespread there, and clades 20A, B and C now dominate global infections and are widespread in Europe [6, 7]. The 20 clades (G clades by GISAID nomenclature) have emerged in Europe in mid-January, and bear the D614G mutation (refers to the mutation in the amino acid sequence; A23403G refers to the nucleotide sequence) in the spike protein which bind the human ACE2 receptor [8]. This mutation has recently been associated with high viral loads and increased infectivity but not with patient hospitalization status [6], although recent reports argue that this variant is related to COVID-19 mortality [9, 10]. Additional mutations within the SARS-CoV-2 genome are being monitored as potential emerging-clades (e.g. C18877T emerging from clade 20, C13730T emerging from clade 19) via Nextstrain's global genomic epidemiology analysis [5] and may become a major clade once they reach sufficient global frequency/spread.

SARS-CoV-2 started to spread in Israel in late February through early March 2020, where multiple importation events of SARS-CoV-2 into Israel from countries worldwide initiated a rapid outbreak across the country with >88,000 infected individuals and ~700 deaths by August 2020. Prompted by recent escalations in the daily number of infected individuals in Israel, in this study we sequenced 160 SARS-CoV-2 complete genomes from imported and early circulating samples. Along with epidemiological data including country of importation and district of residence and additional Israel-based sequences from the same time frame available in GISAID, we thoroughly investigated mutation patterns to characterize the origins of viral evolution and spread patterns of SARS-CoV-2 in Israel.

## Materials and methods

### Sample collection, nucleic acid extraction and viral genome detection by real-time PCR

Starting with the first imported cases into Israel in February and until mid-March 2020, all individuals entering Israel suspected to have contracted SARS-CoV-2 were exclusively diagnosed in Israel's Central Virology Laboratory (ICVL) via real-time PCR from nasal- pharyngeal samples. Samples to be sequenced in this study included all such imported cases (n = 59) and circulating cases, from individuals identified as SARS-CoV-2 positive between mid-March and April (n = 101). Additional 211 sequences from Israel from the same time frame were downloaded from GISAID and included in the analyses.

Viral genomes were extracted from 200 μL respiratory samples with the MagNA PURE 96 (Roche, Mannheim, Germany), according to the manufacturer instructions and real time PCR (RT-PCR) reactions using primers corresponding to the SARS-CoV-2 envelope (E) gene were performed as previously described [11]. All samples were tested for the human RNAseP gene, which served as a housekeeping gene. The RT- PCR reactions were performed in 25 μL Ambion Ag-Path Master Mix (Life Technologies, Carlsbad, CA, USA) using TaqMan Chemistry on the ABI 7500 instrument. Nucleic extraction samples from SARS-CoV-2 positive samples were taken for further molecular analysis.

**Ethics statement.** The study has been approved by the Sheba Medical Center Helsinki committee. This is a retrospective study of archived samples, where sample names were anonymized; institutional Helsinki committee waived the requirement for informed consent.

## Specific amplification of SARS-CoV-2 from clinical samples

RNA in extracted nucleic acids was reverse transcribed to single strand cDNA using Super-Script IV (ThermoFisher Scientific, Waltham, MA, USA) as per manufacturer's instructions. SARS-CoV-2 specific primers designed to capture SARS-CoV-2 whole genome (version 1—total 218 primers, divided into two primer pools designed by Josh Quick from ARTIC Network) were used to generate double strand cDNA and amplify it via PCR using Q5 Hot Start DNA Polymerase (NEB) [12]. Briefly, each sample underwent two PCR reactions with primer pool 1 or 2 and 5X Q5 reaction buffer, 19 mM dNTPs and nuclease-free water. Resulting DNA was combined and quantified with Qubit dsDNA BR Assay kit (ThermoFisher Scientific) as per manufacturer's instructions and 1ng of amplicon DNA in 5 μL per sample was taken into library preparation.

## Library preparation and sequencing

Libraries were prepared using NexteraXT library preparation kit and NexteraXT index kit V2 as per manufacturer's instructions (Illumina, San Diego, CA, USA). Libraries were purified with AMPure XP magnetic beads (Beckman Coulter, Brea, CA, USA) and library concentration was measured by Qubit dsDNA HS Assay kit (Thermo Fisher Scientific, Waltham, MA, USA). Library validation and mean fragment size was determined by TapeStation 4200 via DNA HS D1000 kit (Agilent, Santa Clara, CA, USA). The mean fragment size was ~400 bp, as expected. The library mean fragment size and concentration molarity was calculated and each library was diluted to 4 nM. Libraries were pooled, denatured and diluted to 10pM and sequenced on MiSeq with V3 2X300 bp run kit (Illumina). Sequences are available in GISAID: EPI_ISL_474958 –EPI_ISL_475025, EPI_ISL_649060—EPI_ISL_649107.

## Bioinformatics analyses

Fastq files were subjected to quality control using FastQC (www.bioinformatics.babraham.ac. uk/ projects/fastqc/) and MultiQC [13] and low-quality sequences were filtered using trimmomatic [14]. To obtain a consensus sequence per sample, paired-end fastq files were combined for each sample via Unix cat command. SARS-CoV-2 reference genome was downloaded from the national center for biotechnology information (NCBI) (NC_045512.2) and indexed using Burrows-Wheeler aligner (BWA) [15]. Combined fastq files were mapped to the indexed reference genome using BWA mem [15]. SAMtools suite [16] was used to convert sam to bam files, remove duplicates and filter unmapped reads. Bam files were sorted, indexed and subjected to quality control using SAMtools suite. Coverage and depth of sequencing was calculated from sorted bam files using a custom python script. A consensus sequence was constructed for each sample using SAMtools mpileup and bcf tools [17] and converted to a fasta file using seqtk (https://github.com/lh3/seqtk). Resulting consensus sequences were further analyzed together with additional sequences identified in Israel from late March to late April (n = 211) available in GISAID [4]. Using Augur pipeline [5], sequences were aligned to SARS-CoV-2 reference genome (NC_045512.2) using MAFFT [18], and a time-resolved phylogenetic tree was constructed with IQ-Tree [19] and TreeTime [20] under the GTR substitution model and visualized with auspice [5]. Clade nomenclature was attained from Nextstrain [5].

Additional bioinformatic analyses such as translation from nucleotide to amino acid sequences, comparison of differences across sequences and sample clustering were carried out using R and Bioconductor packages Seqinr [21], HDMD (https://CRAN.R-project.org/package=HDMD) and ggplot2 [22]. Classification to amino acid groups was set according to physiochemical attributes determined by Atchley et al. [23].

## Results

### SARS-CoV-2 genomic epidemiology of imported and early circulating viruses

On February 21, 2020, two Israeli citizens infected with SARS-CoV-2 from the Diamond Princess cruise ship anchoring in Japan were brought to designated SARS-CoV-2 quarantine facilities in Israel. The first non-controlled imported case of SARS-CoV-2 into Israel from Europe (Italy) was diagnosed in February 27, 2020, followed by additional importations, mostly from other European countries but also from countries worldwide until early March, when air traffic was largely suspended. At that time, SARS-CoV-2 suspected individuals were exclusively diagnosed by the central virology laboratory, such that all epidemiologically-verified importations were recorded and samples were retained. Here, we sequenced complete SARS-CoV-2 genomes from all imported cases identified in late February to mid-March and from circulating viruses from individuals diagnosed between mid-March and April. Results were analyzed together with additional sequences identified in Israel from late March to late April available in GISAID [4]. A phylogenetic tree of the imported and circulating cases, shows that importation events from Europe, United States, Asia and Africa (Egypt) in late February to mid-March led to multiple transmission chains in Israel (Fig 1A). All districts in Israel were affected, with the highest number of importations occurring into the Central and Tel-Aviv districts (Fig 1B).

### SARS-CoV-2 imported and circulating clades

To define imported and circulating clades in Israel, we applied the Nextstrain nomenclature (https://github.com/nextstrain/ncov/blob/master/defaults/clades.tsv), that includes the originating clade 19A and its derivation 19B, and the emerging clades 20 and its derivations 20B and 20C with the spike mutation D614G [6] that had widely spread in Europe since mid-February [5]. All five clades were imported into Israel during late February to mid-March (Fig 2A, n = 59). Clades 19A and 19B constituted 40% of the clades imported into Israel with relatively equal representation (~20% each). Clade 19A included the two Diamond Princess samples imported from Japan, and clade 19B was almost exclusively imported from Spain (11/12 of 19B importations) (Fig 2B). Clade 20 constituted 60% of imported cases and included the first importation from Italy (clade 20B) (Fig 2B). Clade 20C, which was equally represented as clades 20A and 20B in the imported population became the dominant circulating clade in Israel (51%), whereas clade 19B diminished in the circulating population (Fig 2C). Within Israel, the Jerusalem and Tel Aviv districts had the highest SARS-CoV-2 incidence and the Haifa district the lowest during the early spread. Clade 20, specifically 20C, was the dominant clade in most districts (Fig 2C).

### SARS-CoV-2 mutation patterns

To further explore patterns in viral evolution, we identified positions along the SARS-CoV-2 genome that were frequently altered across the Israeli sequences compared to the reference genome. Correlations of these positions revealed novel positions that were altered in the Israeli sequences, in addition to the known clade-defining positions (Fig 3A). The novel positions were associated with defined clades via Nextstrain auspice visualization tool [5]. Clusters of

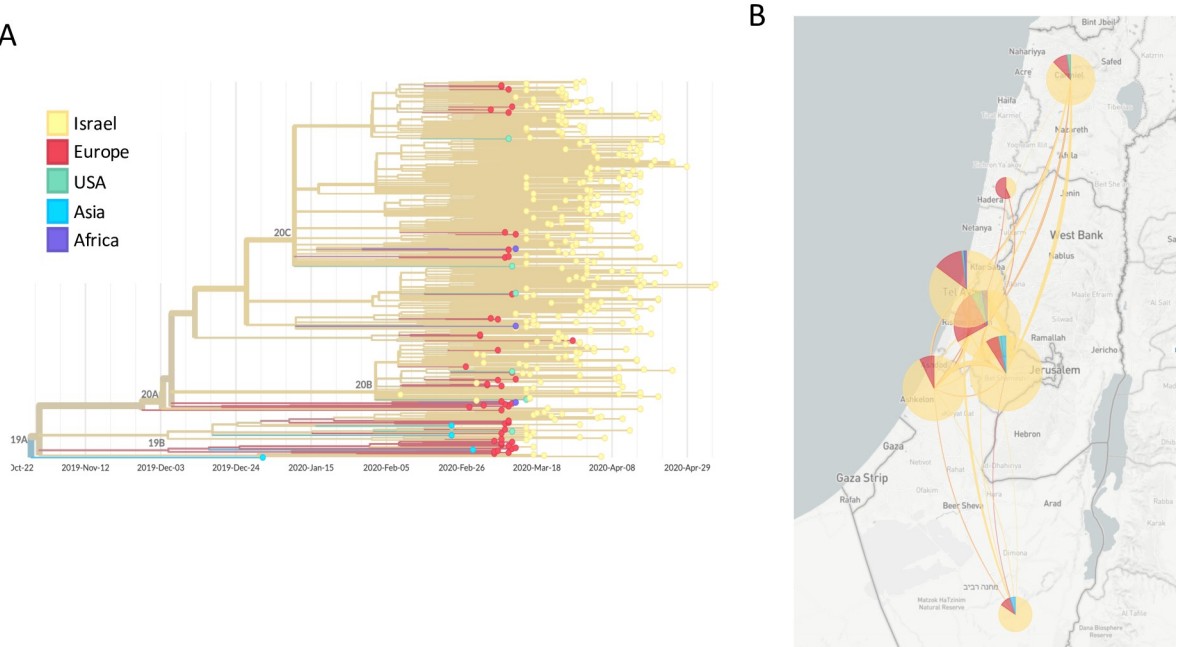

**Fig 1. SARS-CoV-2 genomic epidemiology of samples imported and circulating in Israel. (A)** time-resolved phylogenetic tree representing 372 imported and early circulating samples sequenced in Israel. Samples are colored according to their origin: Israel circulating samples in yellow, and imported samples from Europe, USA, Asia (China-reference sequence, Japan) and Africa (Egypt) in red, green, blue and purple, respectively. SARS-CoV-2 clades are noted by each relevant branch. **(B)** Distribution of imported and circulating samples across districts in Israel.

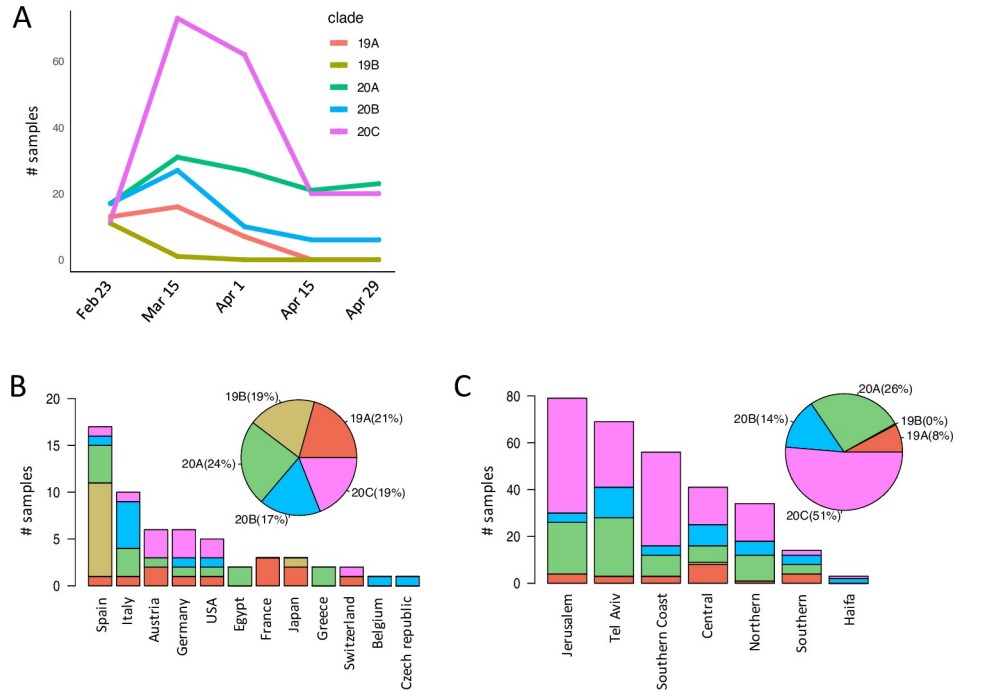

**Fig 2. SARS-CoV-2 imported and circulating clades. (A)** Distribution of SARS-CoV-2 clades from first diagnosed sample in late February through early circulation in Israel. **(B)** Distribution and frequency of clades in imported samples in late February to mid-March, by country of origin. **(C)** Distribution and frequency of clades in early circulating samples (mid-March to late April), by district.

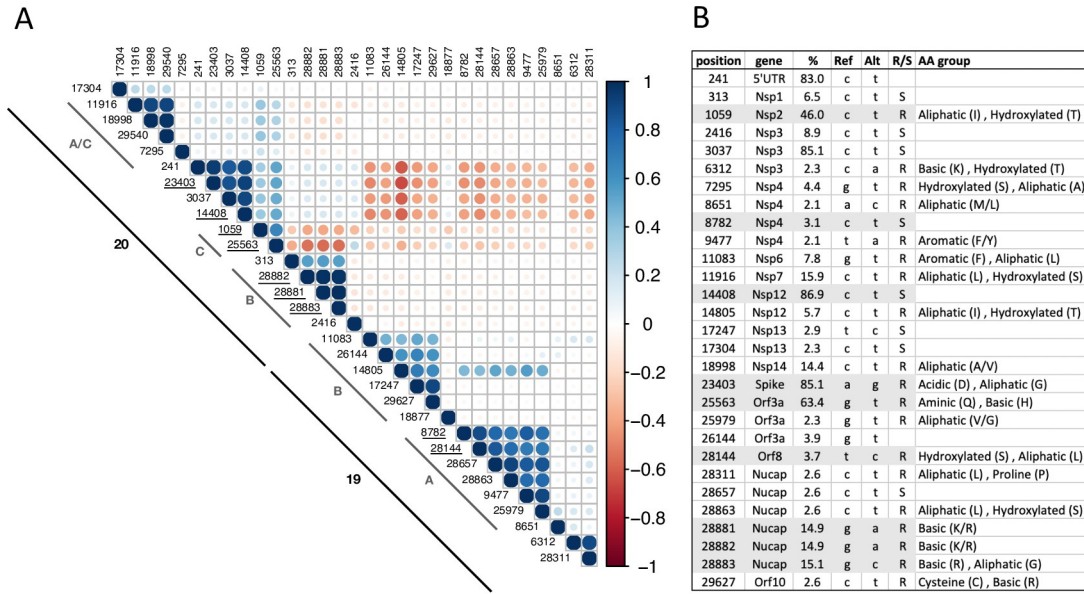

**Fig 3. SARS-CoV-2 frequently observed mutations.** 29 mutations along the SARS-CoV-2 genome occurred in >2% of the 372 Israeli sequences. **(A)** Correlation table of the frequently observed mutations. Positive/negative correlations are denoted in blue/ red respectively. Known clade-defining mutations are underlined and clade association is noted. **(B)** Listed for each frequently observed mutation its position, gene, % frequency in Israeli sequences, nucleotide substitution, whether it's an R or S mutation, and in case of an R mutation, the originating and altered AA group. Known clade-defining mutations are highlighted in grey.

positive correlations were observed between mutated positions within each of the 19 and 20 clades, whereas negative correlations were observed between mutated positions associated with clades 19 and positions in clade 20, suggesting distinct linkage of these positions to either clade. Interestingly, negative correlations were observed between the clade 20B mutated positions (313, 28881, 28882, 28883) and some of the positions in clades A/C (e.g. 1059, 25563, 11916), which may hint that the clade B positions are strongly linked to one another (Fig 3A). Visualization with Nextstrain global analysis (https://nextstrain.org/ncov/global) showed that these mutated positions are not specifically unique to Israel and were observed in several SARS-CoV-2 genome sequences worldwide. To assess the impact of all these alterations, the resultant amino acid (AA) substitutions were classified into silent (S) or replacement (R), and in the latter case, the change in the physiochemical attributes of the AA (classified by Atchley [23]) was also assessed. R mutations were observed in higher frequency (20/29 mutations) compared to S (9/29), most of which led to a change in the AA attribute group (12/20 R mutations). Many of the AA group exchanges involved a change between the aliphatic (non-polar, hydrophobic) and hydroxylated (polar, uncharged) AA groups. Finally, over half of the mutations observed (15/29) were C-to-T, suggesting viral restriction by host APOBEC mechanism in these positions, as previously observed [24].

## Discussion

Since its first importation into Israel late February 2020, SARS-CoV-2 had expeditiously spread in Israel. The first importations occurred from Japan and Europe (Italy), however the spread in the population is more likely to have been initiated by first importations from Europe, as the importations from Japan (Diamond Princess passengers) were planned and controlled in specialized treatment facilities. Sequencing and analyses of SARS-CoV-2 complete genomes from imported and circulating samples revealed that although several clades

were initially imported into Israel in late February to mid-March, clade 20 quickly became dominant, similar to observations across Europe. Clade 20 (including 20A, B and C), also known as clade G by GISAID nomenclature [4], is an emerging clade that has gained prominence in Europe in early March followed by expansion into North America and Asia, where its hallmark mutation, D614G (a23403g in the nucleotide sequence), has been recently shown to increase infectivity [6]. Specifically, clade 20C, a dominant clade in North America [5], Denmark and Finland [7], that was observed in 51% of circulating samples in Israel, may have been reinforced in gaining prominence in Israel by additional importations from the United States in late March, in addition to its naturally higher infectivity compared to clade 19.

Frequently mutated positions were identified in the Israeli samples, some of which correlated with known clade-defining mutations and observed also in sequences worldwide. Most of these mutations were R mutations that caused a change in the AA attribute group, which have a greater chance to affect the protein. It is important to closely observe the emerging mutated positions throughout this continuous pandemic as some may gain evolutionary advantage and affect larger portions of the population. This might have an impact on the specificity of diagnostic tests such as RT-PCR and even vaccine design targeting these positions.

SARS-CoV-2 is still spreading in Israel and across the globe. Surveillance of SARS-CoV-2 genomes is crucial for understanding its evolution and spread patterns and may aid in decision making concerning public health issues.

## Author Contributions

**Conceptualization:** Neta S. Zuckerman, Orna Mor, Michal Mandelboim.

**Data curation:** Neta S. Zuckerman, Orna Mor, Michal Mandelboim.

**Formal analysis:** Neta S. Zuckerman.

**Methodology:** Neta S. Zuckerman, Efrat Bucris, Yaron Drori, Oran Erster.

**Project administration:** Neta S. Zuckerman, Michal Mandelboim.

**Resources:** Orna Mor, Michal Mandelboim.

**Supervision:** Neta S. Zuckerman, Efrat Bucris, Michal Mandelboim.

**Visualization:** Neta S. Zuckerman.

**Writing – original draft:** Neta S. Zuckerman.

**Writing – review & editing:** Neta S. Zuckerman, Danit Sofer, Rakefet Pando, Ella Mendelson, Orna Mor, Michal Mandelboim.

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
