## [Decision Letter · Decision Letter 0]

9 Nov 2020

PONE-D-20-27397

Genomic epidemiology of SARS-CoV-2 importation and early circulation in Israel

PLOS ONE

Dear Dr. Zuckerman,

Thank you for submitting your manuscript to PLOS ONE. After careful consideration, we feel that it has merit but does not fully meet PLOS ONE’s publication criteria as it currently stands. Therefore, we invite you to submit a revised version of the manuscript that addresses the points raised during the review process.

We look forward to receiving your revised manuscript.

Kind regards,

Ahmed S. Abdel-Moneim, Ph.D.

Academic Editor

PLOS ONE

Journal Requirements:

4. We note that Figure 1 in your submission contain map images which may be copyrighted. All PLOS content is published under the Creative Commons Attribution License (CC BY 4.0), which means that the manuscript, images, and Supporting Information files will be freely available online, and any third party is permitted to access, download, copy, distribute, and use these materials in any way, even commercially, with proper attribution. For these reasons, we cannot publish previously copyrighted maps or satellite images created using proprietary data, such as Google software (Google Maps, Street View, and Earth). For more information, see our copyright guidelines: http://journals.plos.org/plosone/s/licenses-and-copyright.

4.1.    You may seek permission from the original copyright holder of Figure 1 to publish the content specifically under the CC BY 4.0 license. 

4.2.    If you are unable to obtain permission from the original copyright holder to publish these figures under the CC BY 4.0 license or if the copyright holder’s requirements are incompatible with the CC BY 4.0 license, please either i) remove the figure or ii) supply a replacement figure that complies with the CC BY 4.0 license. Please check copyright information on all replacement figures and update the figure caption with source information. If applicable, please specify in the figure caption text when a figure is similar but not identical to the original image and is therefore for illustrative purposes only.

Reviewers' comments:

Reviewer's Responses to Questions

**Comments to the Author**

1. Is the manuscript technically sound, and do the data support the conclusions?

Reviewer #1: Yes

Reviewer #2: Yes

2. Has the statistical analysis been performed appropriately and rigorously? 

Reviewer #1: Yes

Reviewer #2: Yes

3. Have the authors made all data underlying the findings in their manuscript fully available?

Reviewer #1: Yes

Reviewer #2: Yes

4. Is the manuscript presented in an intelligible fashion and written in standard English?

Reviewer #1: Yes

Reviewer #2: Yes

5. Review Comments to the Author

Reviewer #1: The paper entitled "Genomic epidemiology of SARS-CoV-2 importation and early circulation in Israel" presents the molecular epidemiology of SARS-CoV-2 cases over time in selected portions of Israel. Overall, the methods and analysis conducted are fairly straightforward and properly used. The findings add to the growing picture of different variant emergence globally, and would be of interest to the field.

However, the figures of the paper are cut off, so it is difficult to completely analyze the findings of the paper. I also believe that the authors could expand a little more in the discussion in terms of comparison of the clade trends observed in the isolates they analyze with other countries, especially those near Israel. Maybe the authors have included it, but a figure more directly associating clade prevalence as a function of time course of the outbreak in Israel would be of value. Finally , I think some minor typos and odd phrasing in places can be corrected, though this would normally be done by a copy editor as they are minor. Overall, I think this would normally be something I would recommend "Minor Revisions" for, but I would encourage the authors to resubmit/clear the figures in full form (not cut off) with the Editor.

Reviewer #2: the paper is clearly written and well organized , the results and figures are comprehensive and helpful, please consider the highlights in the attached file for the minor reversions and justifications for questions.

finally: I have a question about the period of sampling; why you did not conduct your research on samples till (August for example) ? to cover wide range of samples to be more comprehensive about the situation.

6. PLOS authors have the option to publish the peer review history of their article (what does this mean?). If published, this will include your full peer review and any attached files.

Reviewer #1: No

Reviewer #2: **Yes: **Mohamed Samy Abousenna

---

## [Author Response · Author response to Decision Letter 0]

12 Nov 2020

Reviewer #1: The paper entitled "Genomic epidemiology of SARS-CoV-2 importation and early circulation in Israel" presents the molecular epidemiology of SARS-CoV-2 cases over time in selected portions of Israel. Overall, the methods and analysis conducted are fairly straightforward and properly used. The findings add to the growing picture of different variant emergence globally, and would be of interest to the field.

However, the figures of the paper are cut off, so it is difficult to completely analyze the findings of the paper.

 The figures will be re-submitted properly.

I also believe that the authors could expand a little more in the discussion in terms of comparison of the clade trends observed in the isolates they analyze with other countries, especially those near Israel. 

 We thank the reviewer for this comment. We added information regarding dominant clades similar to Israel from additional European countries (line #263). Europe and the USA are relevant to Israel in terms of viral transmission due to travel patterns, more than countries bordering with Israel, and this is the reason those were included. In addition, the paucity of data available for countries near Israel does not allow drawing clear conclusions. 

Maybe the authors have included it, but a figure more directly associating clade prevalence as a function of time course of the outbreak in Israel would be of value. 

 This information is shown in Figure 2A.

Finally, I think some minor typos and odd phrasing in places can be corrected, though this would normally be done by a copy editor as they are minor. 

 We have thoroughly gone over the manuscript again to correct any typos and phrasing.

Overall, I think this would normally be something I would recommend "Minor Revisions" for, but I would encourage the authors to resubmit/clear the figures in full form (not cut off) with the Editor.

Reviewer #2: the paper is clearly written and well organized, the results and figures are comprehensive and helpful, please consider the highlights in the attached file for the minor reversions and justifications for questions.

 We thank the reviewer for these comments. Below are our answers to the comments highlighted in the manuscript:

1. Page #1:

title – changed to “Genomic variation and epidemiology of SARS-CoV-2 importation and early circulation in Israel” as the reviewer suggested.

Author roles:

Conceptualization, N.S.Z and M.M; Methodology, N.S.Z, E.B, Y.D and O.E; Validation, E.B and O.E; Formal analysis, N.S.Z, E.B and Y.D; Investigation, N.S.Z and E.B; Data curation, N.S.Z, R.P and M.M; Writing – original draft, N.S.Z; Writing – review and editing, N.S.Z, E.M, D.S, O.M and M.M; Visualization, N.S.Z; Supervision, N.S.Z, E.M, O.M and M.M; Project administration, N.S.Z, O.M and M.M.

2. Page #2:

Abstract – the number of samples mentioned indicates the samples that were sequenced by us and described in the manuscript. Other samples included are Israeli-based samples taken from GISAID (publically available database). This is mentioned in the abstract: “with additional sequences identified Israel available in public databases”.

In addition, we added the number of sequences we utilized from GISAID in the methods section (line 87).

3. Page #4:

Materials and methods: we added the type of samples – nasal-pharyngeal (line 86).

PCR: term was changed to real time PCR (RT-PCR). In our experience, the maximum Ct value that can be sequenced using the methodology described (targeted SARS-CoV-2 sequencing using the Artic primers) is ~35. 

4. Page #7:

Results: all methods-related duplicate information was deleted.

finally: I have a question about the period of sampling; why you did not conduct your research on samples till (August for example)? to cover wide range of samples to be more comprehensive about the situation.

 The purpose of the manuscript, that was written back in June, was to describe the origin of SARS-CoV-2 in Israel and its initial spread (the “first wave”). This first wave had declined in May 2020, and a “second” wave started in July and is still ongoing. We are currently sequencing samples from that time-period to characterize the second wave.

---

## [Decision Letter · Decision Letter 1]

19 Nov 2020

Genomic variation and epidemiology of SARS-CoV-2 importation and early circulation in Israel

PONE-D-20-27397R1

Dear Dr. Zuckerman,

We’re pleased to inform you that your manuscript has been judged scientifically suitable for publication and will be formally accepted for publication once it meets all outstanding technical requirements.

Kind regards,

Yury E Khudyakov, PhD

Academic Editor

PLOS ONE

Additional Editor Comments (optional):

Reviewers' comments:

Reviewer's Responses to Questions

**Comments to the Author**

1. If the authors have adequately addressed your comments raised in a previous round of review and you feel that this manuscript is now acceptable for publication, you may indicate that here to bypass the “Comments to the Author” section, enter your conflict of interest statement in the “Confidential to Editor” section, and submit your "Accept" recommendation.

Reviewer #1: All comments have been addressed

Reviewer #2: All comments have been addressed

2. Is the manuscript technically sound, and do the data support the conclusions?

Reviewer #1: Yes

Reviewer #2: Yes

3. Has the statistical analysis been performed appropriately and rigorously? 

Reviewer #1: Yes

Reviewer #2: Yes

4. Have the authors made all data underlying the findings in their manuscript fully available?

Reviewer #1: Yes

Reviewer #2: Yes

5. Is the manuscript presented in an intelligible fashion and written in standard English?

Reviewer #1: Yes

Reviewer #2: Yes

6. Review Comments to the Author

Reviewer #1: I thank the authors for addressing my comments--the manuscript looks great! They pretty much addressed everything I had, though one more read-through for phrasing would be good.

Reviewer #2: Please address the required CT value at materials and methods, as you explained in your response to cooments

7. PLOS authors have the option to publish the peer review history of their article (what does this mean?). If published, this will include your full peer review and any attached files.

Reviewer #1: No

Reviewer #2: **Yes: **Mohamed Samy Abousenna

---

## [Editor Report · Acceptance letter]

18 Mar 2021

PONE-D-20-27397R1 

Genomic variation and epidemiology of SARS-CoV-2 importation and early circulation in Israel 

Dear Dr. Zuckerman:

I'm pleased to inform you that your manuscript has been deemed suitable for publication in PLOS ONE. Congratulations! Your manuscript is now with our production department. 

Kind regards, 

on behalf of

Dr. Yury E Khudyakov 

Academic Editor

PLOS ONE